# Potential of Bioactive Food Components against Gastric Cancer: Insights into Molecular Mechanism and Therapeutic Targets

**DOI:** 10.3390/cancers13184502

**Published:** 2021-09-07

**Authors:** Seog Young Kang, Dongwon Hwang, Soyoung Shin, Jinju Park, Myoungchan Kim, MD. Hasanur Rahman, Md. Ataur Rahman, Seong-Gyu Ko, Bonglee Kim

**Affiliations:** 1College of Korean Medicine, Kyung Hee University, Hoegidong Dongdaemungu, Seoul 05253, Korea; pionasy@khu.ac.kr (S.Y.K.); d.hwang@khu.ac.kr (D.H.); red4103@khu.ac.kr (S.S.); jinjuzz98@khu.ac.kr (J.P.); rahman23@khu.ac.kr (M.A.R.); 2Department of Pathology, College of Korean Medicine, Kyung Hee University, Hoegidong Dongdaemungu, Seoul 05253, Korea; dongoorai@khu.ac.kr; 3Department of Biotechnology and Genetic Engineering, Bangabandhu Sheikh Mujibur Rahman Science and Technology University, Gopalganj 8100, Bangladesh; hasan079@bsmrstu.edu.bd; 4Korean Medicine-Based Drug Repositioning Cancer Research Center, College of Korean Medicine, Kyung Hee University, Hoegidong Dongdaemungu, Seoul 05253, Korea; epiko@khu.ac.kr

**Keywords:** gastric cancer, bioactive food components, autophagy, apoptosis, angiogenesis, metastasis, chemo-resistance

## Abstract

**Simple Summary:**

Recently, it has been found that cancer of the gastrointestinal tract, especially gastric cancer (GC), is the second most leading cause of cancer-related death globally. Extensive research has shown that most epidemiological investigations indicated the increased intake of naturally-occurring bioactive food components could decrease the gastric cancer risk. Several experimental studies have explained that the molecular mechanisms of action to prevent GC comprise induction of apoptosis, inhibition of cell proliferation, suppression of angiogenesis and metastasis, and regulation of autophagy. To provide an updated understanding of relationships between naturally occurring bioactive food components and gastric cancer, this study will be helpful for guiding and preventing gastric cancer by natural bioactive food products.

**Abstract:**

Gastric cancer, also known as stomach cancer, is a cancer that develops from the lining of the stomach. Accumulated evidence and epidemiological studies have indicated that bioactive food components from natural products play an important role in gastric cancer prevention and treatment, although its mechanism of action has not yet been elucidated. Particularly, experimental studies have shown that natural bioactive food products display a protective effect against gastric cancer via numerous molecular mechanisms, such as suppression of cell metastasis, anti-angiogenesis, inhibition of cell proliferation, induction of apoptosis, and modulation of autophagy. Chemotherapy remains the standard treatment for advanced gastric cancer along with surgery, radiation therapy, hormone therapy, as well as immunotherapy, and its adverse side effects including neutropenia, stomatitis, mucositis, diarrhea, nausea, and emesis are well documented. However, administration of naturally occurring bioactive phytochemical food components could increase the efficacy of gastric chemotherapy and other chemotherapeutic resistance. Additionally, several studies have suggested that bioactive food components with structural stability, potential bioavailability, and powerful bioactivity are important to develop novel treatment strategies for gastric cancer management, which may minimize the adverse effects. Therefore, the purpose of this review is to summarize the potential therapeutic effects of natural bioactive food products on the prevention and treatment of gastric cancer with intensive molecular mechanisms of action, bioavailability, and safety efficacy.

## 1. Introduction

The incidence and mortality of cancer is growing worldwide, with an estimated 19.3 million new cases and 10 million cancer deaths in 2020 [1]. Gastric cancer is the fifth most common neoplasm and the fourth leading cause of cancer death, which has led to over one million new cases and an estimated 769,000 deaths in 2020 [1]. Clinically, to offer pertinent treatment, gastric carcinoma is classified as early or advanced stage [2]. Gastric carcinoma has multiple risk factors: genetics, *Helicobacter pylori* infection, gastric ulcer, gastroesophageal reflux disease (GERD), tobacco, smoking, alcohol, chemical exposure, diet, obesity, and so forth [3,4]. Surgical resection, when possible, offers the best chances of cure for early gastric cancer [5]. Adjuvant or neoadjuvant chemotherapy may be beneficial in increasing the chance of successful resection or in decreasing the rate of recurrence and/or metastasis [6,7,8]. For patients with unresectable advanced gastric cancer, chemotherapy is a common choice. Conventional regimens are mostly based on cytotoxic agents including antimetabolites and platinum-based anticancer drugs. However, these regimens cause severe side effects such as chemotherapy-induced peripheral neuropathy (CIPN), neutropenia, stomatitis, mucositis, diarrhea, nausea, and emesis [9,10]. Moreover, failure of first-line chemotherapy due to resistance is also an obstacle of gastric cancer treatment hampering the novel and effective therapies and imposing significant economic costs to patients [11]. Moreover, exposure to unremovable toxins (not able to be removed or non-releasable), trauma, or infection lead to mutagenic chronic inflammatory responses, which cause dysplasia [12]. Considering gastric cancer, *Helicobacter pylori* infection is a major risk factor for developing deleterious tumor microenvironments [13]. Nuclear factor kappa-B (NF-κB), c-Jun N-terminal kinase (JNK), and signal transducer activator of transcription 3 (STAT3), inflammatory cytokines, tumor necrosis factor (TNF), interleukin (IL)-1/6, tumor-derived cytokines such as fasciclin (Fas) ligand, and vascular endothelial growth factor (VEGF) are major targets of regulation for the prevention and treatment of gastric cancer [14,15,16,17,18]. Therefore, novel drug development against gastric cancer is strongly needed to further improve survival rates of this disease and lower the side effects of conventional therapies. 

Epidemiological studies have shown that natural dietary bioactive food components decrease the risks of gastric cancer [19,20,21,22]. Extensive research was conducted to measure the value of natural products for the prevention and treatment of gastric carcinoma, leading to the discovery of major bioactive phytochemicals with anti-cancer properties, such as quercetin, silymarin, taurine, berberine, curcumin, and so forth [23,24,25,26]. However, few review articles included agents from animal or marine sources, which are also being studied with growing expectation [27,28]. The same goes for traditional medicine, despite their wide use in clinical practice to combat various illnesses including cancer [29,30,31,32]. This review explores various bioactive compounds isolated from biological resources of bioactive food components and traditional medicine in the form of single compounds that show anti-cancer properties closely targeted to gastric cancer. Moreover, the use of bioactive food components could be a promising adjuvant remedy for gastric cancer treatment as well as in developing functional food components and drugs for the treatment and prevention of gastric cancer.

## 2. Methods

While there have been similar reviews highlighting the anti-neoplastic efficacies of bioactive food components, few of them were written with regards to the chemical classification of each bioactive compound. This review is not only a simple compilation of previous in vitro studies testing bioactive food components on gastric cancer but goes as far as to systematically organizing previous works depending on each cancer-related pathway, namely apoptosis, autophagy, metastasis, drug-resistant capability, and more. Literature-based online databases, Google Scholar, Web of Science, PubMed, Google, and Scopus were accessed to collect information on the published articles. As there is currently no golden standard for classifying phytochemicals, we adopted a comprehensive and clear method previously demonstrated in a literature highlighting the efficacies of bioactive food components on gastrointestinal diseases. This will help researchers rule out or select appropriate candidate species of natural bioactive food products for further studies. This review only included studies published from 2014 to 2021.

## 3. Apoptosis-Inducing Natural Bioactive Food Components in Gastric Cancer

Apoptosis is the process of programmed cell death, characterized by distinct morphology: cell shrinking, membrane blebbing, chromatin condensation, and nuclear fragmentation [33,34]. Several bioactive compounds showing apoptosis-inducing effects on gastric cancer cells and animal models are presented in Figure 1 and Table 1. Yang et al. reported that berberine could inhibit the proliferation of SGC-7901 cells and induce apoptosis [35]. In vitro models have demonstrated that cyclovirobuxine D originated from *Buxux microphylla* Richardii. *Radix* (Buxaceae) induced apoptosis in MGC-803 and MKN-28 cells [36]. Expressions of caspase-3, cytochrome c, endonuclease G (Endo G), apoptosis inducing factor (AIF), and Smac/Diablo were upregulated in melittin-treated SGC-7901 cells. Trifolirhizin, a compound isolated from *Sophora flavescens* Aiton Radix (Fabaceae), demonstrated apoptotic activity both in vitro and in vivo [37]. Trifolirhizin induced apoptosis of MKN-45 cells in vitro via EGFR-MAPK pathways and triggered G2/M phase cell cycle arrest by impacting the CDC2/Cyclin B complex. Qian et al. discovered that ginsenoside-Rh2 originated from *Panax ginseng* C.A. Mey, Radix (Araliaceae) inhibits proliferation and induces apoptosis of SGC-7901 cells by induction of the Bcl-like protein 4 (Bax) to Bcl-2 (Bax/Bcl-2) ratio [38].

Tanshinone IIA, originated from *Salviae miltiorrhiza* Bunge. *Radix* (Lamiaceae), suppressed AGS gastric tumor cells via activation of tumor necrosis factor-alpha (TNF-α), Fas, p38, JNK, p53, p21, caspase-3, and caspase-8 and inhibition of ERK [39]. [6]-gingerol treatment for 24 h to AGS cells generated ROS and decreased ΔΨm, leading to induction of apoptosis. Perturbations of ΔΨm were associated with deregulation of the Bax/Bcl-2 ratio at the protein level, which led to the upregulation of cytochrome c and triggered the caspase cascade. 2,7-dihydroxy-3-methylanthraquinone (DDMN), a flavone isolated from *Hedyotis diffusa* Willd. *Herba*, induced caspase-dependent apoptosis of SGC-7901 gastric cancer cells [40]. 6,7,30-trimethoxy-3,5,40-trihydroxy flavone (TTF), from *Chrysosplenium nudicaule* Ledeb. Herba, is a well-known traditional Chinese medicine for digestive diseases [41], which induced apoptosis on SGC-7901 cells. Sun et al. observed that curcumin, isolated from *Curcuma longa* L. *Rhizoma* (Zingiberaceae), induced apoptosis of SGC-7901 and BGC-823 cells by up-regulating microRNA-33b (miR-33b) expression [42]. Esculetin treatment triggered ROS formation, elevated caspase-3/9 activity, and induced poly (ADP-ribose) polymerase (PARP) cleavage [43]. Liu et al. reported that hydroxysafflor yellow A (HSYA) induces apoptosis of BGC-7901 gastric carcinoma cells via activation of the peroxisome proliferator-activated receptor gamma (PPARγ) signal through elevation of PPARγ and caspase-3 [44]. Kurarinone synergized TRAIL-induced apoptosis against gastric cancer cell line SGC-7901 [45]. Licochalcone A (LicA), a flavonoid isolated from licorice root, elucidated apoptosis by blocking the Akt signaling pathway and reducing hexokinase 2 (HK2) expression in MKN45 cells [46]. Curcuzedoalide, sesquiterpene bioactive components of *Curcuma zedoaria* Roscoe *Rhizoma* (Zingiberaceae), induced mitochondrial apoptosis induction with cleavage of PARP as well as caspase-8, caspase-9, and caspase-3 in AGS cells [47]. Thymol showed cytotoxicity on AGS cancer cells via the intrinsic mitochondrial pathway via upregulation of Bax and PARP expression, and also promoted cleavage of caspase-7, caspase-8, and caspase-9 and downregulated ΔΨm [48].

The apoptotic ability of ophiopogonin B, the active compound isolated from *Ophiopogon japonicus Radix*, against SGC-7901 cells were suspected to be relevant with the JNK 1/2 and ERK1/2 signaling pathways through upregulation of active caspase-3 and modulation of Bax/Bcl-2 expression [49]. It has been found that phloretin, a plant-derived natural bioactive product, is an important molecule for the treatment of AGS gastric cancer via expression of Bax and was increased in dose-dependently while the expression of Bcl-2 decreased [50]. Podophyllotoxin, isolated from *Linum album* Kotschy (Linaceae), induced apoptosis and downregulated zinc finger protein 703 oncogene expression [51]. Grifolin, isolated from the mushroom *Albatrellus confluens* (Alb. and Schwein) Kotl. and Pouzar (Albatrellaceae), inhibited growth and invasion of gastric cancer cells by inducing apoptosis and suppressing the ERK1/2 pathway [52]. Tsai et al. found that 7-acetylsinumaximol B (7-AB), discovered from *Sinularia sandensis* (Alcyoniidae), showed anti-proliferative effects through apoptosis against human gastric carcinoma NCI-N87 cells via the expression of Bad, Bcl-like protein 11 (Bim), Bax, and cytochrome c, and it decreased the expression levels of phosphorylated Bad (p-Bad), myeloid cell lukemia-1 (Mcl-1), Bcl-xL, and Bcl-2 proteins. [53] Crosolic acid, isolated from *Actinidia valvata* Dunn. *Radix* (Actinidiaceae), was reported to inhibit proliferation of BGC-823 cells by downregulating the NF-κB pathway [54]. Crosolic acid inhibited phosphorylation of nuclear factor kappa B-alpha (IκBα), expression of p65, and nuclear translocation and DNA-binding activity of NF-κB. Deacetylisovaltratum, derived from *Patrinia heterophylla* Bunge, induced mitochondrial and caspase-dependent apoptosis in AGS and HGC-27 cells [55]. Li et al. demonstrated that elemene, a sesquiterpenoid mixture isolated from a traditional herbal medicine, *Curcuma zedoaria* Roscoe *Rhizoma* (Zingiberaceae), countered gastric cancer via regulation of the ERK 1/2 signaling pathway [56]. Liao et al. reported that n-butylidenephthalide (BP), a bioactive compound of *Angelica Sinensis* Diels *Radix*, activated the intrinsic apoptotic pathway of human gastric cancer cells AGS, NCI-N87, and TSGH-9201 [57]. Paeonol treatment inhibited proliferation, invasion, migration, and induced apoptosis against BGC823 cells. The protein expression of matrix metalloproteinase (MMP)-2 and MMP-9 were attenuated in a concentration-dependent manner by paeonol [58]. Pseudolaric acid B, isolated from *Pseudolarix amabilis*, commonly called golden larch, inhibited cell proliferation and induced apoptosis of the multidrug-resistant SGC-7901/ADR gastric cancer cell line [59]. 

Thymol is a phenolic compound isolated from *Thymus quinquecostatus* Celak. (Lamiaceae) that possesses anti-inflammatory, anticancer, antibacterial, and more biological efficacies [48]. The anticancer potencies of toosendanin (TSN), a triterpenoid found in *Melia toosendan* Sieb et Zucc *Cortex et Fructus* (Meliaceae), was discussed in two studies. Wang et al. found that SGC-7901 cells treated with toosendanin (TSN) increased early apoptosis [60]. TSN inactivated the β-catenin pathway in SGC-7901 cells and subsequently induced apoptosis following facilitation of microRNA 200a [60]. It has been reported that peptic oligosaccharide, separated from *Solanum lycopersicum* L. (Solanaceae), induced apoptosis by suppressing galectin-3 expressions [61]. Additionally, several natural bioactive products retarded tumor growth in animal models, as presented in Table 2. Wu et al. revealed that phenolic alkaloids of *Menispermum dauricum* induced apoptosis and suppressed gastric tumor growth by inducing apoptosis and inhibiting oncogenic Kirsten Rat sarcoma viral oncogene homolog (K-RAS) expression [62]. When BALB/C mice grafted with MFC mouse gastric cancer cells were treated with curcumin solution every day for 60 days, expressions of interferon gamma (IFN-γ), tumor necrosis factor-alpha (TNF-α), granzyme B, and perforin were upregulated, while differentiated embryonic chondrocyte gene 1 (DEC1), hypoxia-inducible factor-1 alpha (HIF-1α), STAT3, and VEGF expression were downregulated in the experimental group [63]. When MKN45-treated BALB/ca mice were treated with LicA, tumor growth was significantly inhibited in contrast to the vehicle group without LicA treatment [46]. Elemene retarded tumor growth in nude mice and showed better efficacy when synergized with PD98059 [56]. In a xenograft mouse model, mice treated with grifolin survived for a longer period compared to the control group [52].

**Table 1 cancers-13-04502-t001:** Apoptosis-inducing bioactive food components in vitro. (↑ increase, ↓ decrease).

Classification	Compound	Source	Experimental Model	Dose; Duration	Efficacy	Mechanism	References
Alkaloids	Berberine	(family: Ranunculaceae) *Coptidis japonica* Makino *Rhizoma*	SGC-7901	5, 10, 20 µM; 24, 48 h	Induction of apoptosis		[35]
Alkaloids	Cyclovirobuxine D	(family: Buxaceae) *Buxus microphylla* Richardii *Radix*	MGC-803, MKN-28	30, 60, 120 μM/L; 48 h	Induction of apoptosis	↑c-caspase-3, Bax ↓Bcl-2	[36]
Alkaloids	GFG-3a	(family: Meripilaceae) *Grifola frondose* (Diks.) Gray *Mycelia*	SGC-7901	100, 200 μg/mL; 24, 48 h	Induction of apoptosis	↑RBBP4, caspase-3, -8, p53, Bax, Bad ↓RUVBL, NPM, Bcl-2, Bcl-xL, PI3K, Akt1	[64]
Alkaloids	Melittin	(family: Apidae) *Apis cerena* Fabricius *venom*	SGC-7901	4 μg/mL; 1, 2, 4 h	Induction of apoptosis	↑caspase-3, cyt c, Endo G, AIF, Smac/Diablo, ROS ↓ΔΨm	[65]
Alkaloids, Terpenoids	Berberine, d-Limonene	(1) (family: Ranunculaceae) *Coptidis japonica* Makino *Rhizoma* (2) (family: Rutaceae) *Evodiae rutaecarpa* Bentham. *Fructus*	MGC-803	(1) 20 µM; 24, 36, 48 h (2) 80 µM; 24, 36, 48 h	Induction of apoptosis	↑ROS, caspase-3 ↑ΔΨm, Bcl-2	[66]
Flavonoids	Trifolirhizin	(family: Fabaceae) *Sophora flavescens* Aiton *Radix*	MKN-45	20, 30, 40 µg/mL; 48 h	Induction of apoptosis	↑caspase-9, -3, c-PARP, p53, p38 ↓EGFR, CDC2, cyclin B, ΔΨm	[37]
Phytosterols	Ginsenoside-Rh2	(family: Araliaceae) *Panax ginseng* C.A. Mey *Radix*	SGC-7901	5, 10, 20 μg/mL; 24, 48 h	Induction of apoptosis	↑Bax ↓Bcl-2	[38]
Phytosterols	Periplocin	(family: Apocynaceae) *Periplocae sepium* Bunge.	SGC-7901, MGC-803, BGC-823	50, 100, 200 ng/mL; 24, 48 h	Induction of apoptosis	↑Mcl-1, c-caspase-3, EGR 1 ↓pro-Bid, p-ERK 1/2	[67]
Phytosterols	Tanshinone IIA	(family: Lamiaceae) *Salviae miltiorrhiza* Bunge. *Radix*	AGS	2.0, 3.7, 5.5 µg/mL; 24, 48 h	Induction of apoptosis	↑TNF-α, Fas, p-p38, p-JNK, p53, p21, caspase-8, -3 ↓p-ERK, CDC2, cyclin A, cyclin B1	[39]
Polyphenols	[6]-Gingerol	(family: Zingiberaceae) *Zingiber officinale* Roscoe *Rhizoma*	AGS	100, 250 µM; 24 h	Induction of apoptosis	↑cyt c, Bax ↓Bcl-2	[68]
Polyphenols	2,7-dihydroxy-3-methylanthraquinone (DDMN)	(family: Rubiaceae) *Hedyotis diffusa* Wild *Herba*	SGC-7901	10, 20, 40 µM; 48 h	Inhibition of proliferation	↑Bax, Bad, caspase-3, -9, cyt c ↓Bcl-xL, Bcl-2	[40]
Polyphenols	6, 7, 30-trimethoxy-3, 5, 40 -trihydroxy flavone (TTF)	(family: Saxifragaceae) *Chrysosplenium nudicaule* Ledeb *Herba*	SGC-7901	2, 4, 8, 16, 32 µg/mL; 24, 48, 72 h	Induction of apoptosis	↑endogenous Ca2+/Mg2+ dependent endonuclease	[41]
Polyphenols	Curcumin	(family: Zingiberaceae) *Curcuma longa* L. *Rhizoma*	SGC-7901, BGC-823	5, 10, 15, 20, 40 μM/L; 24 h	Induction of apoptosis	↓XIAP ↑miR-33b	[42]
Polyphenols	Esculetin	(family: Asteraceae) *Artemesia scoparia* Waldst. et Kit, *Artemesia capillaris* Thunb.)) (family: Plumbaginaceae)	SGC-7901, MGC-803, BGC-823	12.5, 25, 50 μM; 24 h	Induction of apoptosis	↑ROS, c-caspase-9, -3, c-PARP, cyt c, Bak, Bax, CypD ↓Bcl-2, Bcl-xL, XIAP	[43]
Polyphenols	Hydroxysafflor Yellow A	(family: Asteraceae) *Carthamus tinctorius* L.	BGC-823	100 µM; 48 h	Induction of apoptosis	↑caspase-3, PPARγ	[44]
Polyphenols	Kurarinone	(family: Fabaceae) *Sophora flavescens* Aiton *Radix*	SGC-7901	5 μM; 24 h	Enhancement ofTRAIL-induced apoptosis	↓Mcl-1, c-FLIP, p-STAT3	[45]
Polyphenols	Licochalcone A	(family: Fabaceae) *Glycyrrhiza glabra* L. *Root*	MKN-45, SGC-7901	15, 30, 60 µM; 24 h	Inhibition of cell proliferation and tumor glycolysis	↑c-caspase-3, c-PARP ↓Bcl-2, Mcl-1, HK2, p-Akt, p-ERK1/2, p-S6, p-GSK3β	[46]
Polyphenols	Ophiopogonin B	(family: Asparagaceae) *Ophiopogon japonicus* Thunb *Root*	SGC-7901	5, 10, 20 μM	Induction of apoptosis	↑ROS, Bax, caspase-3 ↓p-ERK 1/2, p-JNK 1/2, ΔΨm, Bcl-2	[49]
Polyphenols	Phloretin		AGS	4, 8, 16µM; 24 h	Induction of apoptosis Inhibition of invasion	↑Bax ↓Bcl-2	[50]
Polyphenols	Podophyllotoxin	(family: Linaceae) *Linum album* Kotschy	AGS	200, 400, 600, 800, 1000 µg/mL; 24 h	Induction of apoptosis	↓ZNF703	[51]
Terpenoids	7-Acetylsinumaximol B	(family: Alcyoniidae) *Sinularia sandensis*	NCI-N87	4, 8, 16 µM; 24 h	Induction of apoptosis	↑Bad, Bim, Bax, cyt c ↓p-Bad, Mcl-1, Bcl-xL, Bcl-2	[53]
Terpenoids	Crosolic Acid	(family: Actinidiaceae) *Actinidia valvata* Dunn *Radix*	BGC-823	20, 40, 80 μg/mL; 72 h	Induction of apoptosis	↑Bax, smac, IκBα ↓Fas, Bcl-2, p65, p-IκBα, NF-κB	[54]
Terpenoids	Curcuzedoalide	(family: Zingiberaceae) *Curcuma zedoaria* Roscoe *Rhizoma*	AGS	100, 200 µM; 24 h	Induction of apoptosis	↑c-caspase-8, -9, -3, c-PARP	[47]
Terpenoids	Deacetylisovaltratum	(family: Caprifoliaceae) *Patrinia heterophylla* Bunge.	(1) AGS (2) HGC-27	(1) 4, 8, 16 μM; 24 h (2) 10, 20, 30 μM; 24 h	Induction of apoptosis	↑p21, caspase-3, c-PARP ↓p-STAT3, pro-caspase-9, ΔΨm	[55]
Terpenoids	Elemene	(family: Zingiberaceae) *Curcuma zedoaria* Roscoe *Rhizoma*	BGC-823	20, 40, 80, 160 μg/mL: 24 h	Induction of apoptosis	↑Bax, p-ERK 1/2 ↓Bcl-2	[56]
Terpenoids	Grifolin	(family: Albatrellaceae) *Albatrellus confluens* (Alb. and Schwein.) Kotl. and Pouzar	BGC-823, SGC-7901	10, 50 µM; 48 h	Induction of apoptosis	↑caspase-9, -3, CDKN2 ↓MEK1, MEKK3 MEK5	[52]
Terpenoids	N-butylidenephthalide	(family: Apiaceae) *Angelica Sinensis* Diels *Radix*	AGS	25, 50, 75 µg/mL; 24 h	Induction of apoptosis	↑REDD1 ↓mTOR	[57]
Terpenoids	Paeonol	(family: Paeoniaceae) *Paeonia suffruticosa* Andr *Root bark*, (family: Apocynaceae) *Cynanchum paniculatum* K. Schum *Radix*	BGC-823	0.1, 0.2, 0.4 mg/mL; 24, 48 h	Inhibition of proliferation, invasion, and migration Induction of apoptosis	↓MMP-2, -9	[58]
Terpenoids	Pseudolaric acid B	(family: Pinaceae) *Pseudolarix kaempferi* Gorden *Root bark*	SGC-7901/ADR	5, 10, 20 μM/L; 24 h	Induction of apoptosis	↑p53, Bax ↓P-gp, COX-2, Bcl-2, Bcl-xL	[59]
Terpenoids	Thymol	(family: Lamiaceae) *Thymus quinquecostatus* Celak *Essential oil*	AGS	100, 200, 400 µM; 6, 12, 24 h	Induction of apoptosis	↑Bax, c-PARP, caspase-8, caspase-7, caspase-9 ↓ΔΨm	[48]
Terpenoids	Toosendanin	(family: Meliaceae) *Melia toosendan* Sieb et zucc *Cortex or Fructus*	SGC-7901	0.5, 1 µM; 48 h	Inhibition of invasion, migration and EMT Induction of apoptosis	↑E-cadherin ↓β-catenin	[60]
						↑miR-200a	

**Table 2 cancers-13-04502-t002:** Apoptosis-inducing bioactive food components in vivo. (↑ increase, ↓ decrease).

Classification	Compound	Source	Experimental Model	Dose; Duration	Efficacy	Mechanism	References
Alkaloids	Phenolic alkaloids	(family: Menispermaceae) *Menispermum dauricum* DC. *Rhizoma*	Nude mice/SGC-7901	5, 10, 20 mg/kg/week; 3 weeks	Suppression of tumor growth		[62]
Flavonoids	Trifolirhizin	(family: Fabaceae) *Sophora flavescens* Aiton. *Radix*	BALB/C nude mice/MKN-45	1–3 mg/kg; 3 weeks	Retardation of tumor growth	↑c-caspase-3 ↓ΔΨm	[37]
Polyphenols	2,7-dihydroxy-3-methylanthraquinone (DDMN)	(family: Rubiaceae) *Hedyotis diffusa* Wild. *Herba*	nude mice/SGC-7901	40 mg/kg; 5, 10, 15, 20 days	Inhibition of gastric cancer cell growth	↑Bax, Bad, c-caspase-3, -9, cyt c ↓Bcl-xL, Bcl-2	[40]
Polyphenols	Curcumin	(family: Zingiberaceae) *Curcuma longa* L. *Rhizoma*	BALB/C mice/MFC	20, 40, 60 μM/L/day; 60 days	Inhibition of tumor growth Induction of apoptosis Activation of immune cells	↑IFN-γ, TNF-α, granzyme B, perforin ↓DEC1, HIF-1α, STAT3, VEGF	[63]
Polyphenols	Licochalone A	(family: Fabaceae) *Glycyrrhiza glabra* L. *Radix*	BALB/ca nude mice/MKN-45	10 mg/kg/day; 33 days	Inhibition of tumor growth		[46]
Terpenoids	Elemene	(family: Zingiberaceae) *Curcuma longa* L. *Rhizoma*	BALB/c athymic nude mice/BGC-823	200 mg/kg/day; 15 days	Retardation of tumor growth		[56]
Terpenoids	Grifolin	(family: Albatrellaceae) *Albatrellus confluens* (Alb. and Schwein.) Kotl. and Pouzar	Balb/c nude mice/BGC-823, SGC-7901	15 mg/kg; 2 days	Improvement of survival time		[52]

## 4. Role of Autophagy in Gastric Cancer Treatment Mediated by Natural Bioactive Food Products

Autophagy is a cellular process in which cytoplasmic contents are degraded within the lysosome/vacuole, and the resulting constituents are recycled [69,70]. Autophagy can be classified into macroautophagy, microautophagy, and chaperone-mediated autophagy (CMA) [71]. Among these, macroautophagy, which has been studied the most, is the process of forming autophagosomes that surround organelles and fuse with lysosomes, and natural products modulate autophagy [72,73]. Based on the isolation target, separate kinds of selective autophagy such as mitophagy, pexophagy, and xenophagy can be distinguished [74]. Macroautophagy consists of several sequential steps: initiation, nucleation, elongation, maturation, and fusion with the lysosome [73,75]. Phagosomes originate from omegasomes, subdomains of the ER, and associate with other organelles such as the mitochondria, golgi complex, plasma membrane, recycling endosome, etc., during its development. Four molecules, Unc-51-like kinase 1/2 (ULK1/2), autophagy-related gene 13 (ATG13), family 200-kD interacting protein (FIP200), and Atg101 form the ULK1/2 complex and initiate the process [73]. The mechanistic target of rapamycin complex 1 (mTORC1) is a major inhibitor of the ULK1/2 complex [69,76]. AMP-activated protein kinase (AMPK) inhibits mTORC1 and leads to the activation of the ULK1/2 complex [75]. The ULK1/2 complex phosphorylates the class III phosphatidylinositol-3-kinase (PI3K) vacuole protein sorting 34 (VPS34) complex consisting of VPS15, Beclin-1, and AtG14 complex, which promotes the formation of phosphatidylinositol-3-phosphate (PI3P), which is an essential lipid molecule required for the nucleation step of the phagophore [77]. Atg12 binds with Atg5 and composes a complex with Atg16L. The Atg12-5-16L1 complex lipidates LC3-I into LC3-II [78,79]. LC3-II, considered a marker of autophagy, is essential for phagosome elongation and fusion [80,81]. When the phagosome encloses and becomes a mature autophagosome, it fuses with a lysosome, and degradation and recycling processes follows. Bioactive food compounds were reported to induce autophagy along with apoptosis against gastric cancer cells, as presented in Figure 2.

It has been found that cinnamaldehyde, the bioactive ingredient in *Cinnamomum cassia*, suppressed tumor growth and the migratory and invasive abilities of gastric cancer [82]. Rottlerin, isolated from *Mallotus philipensis* Muell (Euphorbiaceae), induced autophagy and caspase-independent apoptosis against SGC-7901 and MGC-803 cells by downregulating mTOR and S-phase kinase-associated protein 2 (Skp2) [83]. Moreover, treatment of latcripin 1 protein, found in *Lentinula edodes*, activated autophagy of gastric cancer cell lines BGC-823 and SGC-7901 with autophagosome formation via the alteration of LC3-I into LC3-II expression [84]. Oxyresveratrol, found in grape, has been found to accumulate ROS production and initiated autophagic and apoptotic cell death via the FOXO-caspase-3 pathway [85,86]. Kaempferol, a natural bioactive flavonoid, induced autophagic cell death in gastric cancer via IRE1/JNK/CHOP and AMPK/ULK1 pathways [87]. It has demonstrated cytotoxic activity on AGS, MKN-45, and KATO-III human gastric cancer cells via induction of caspase activation and autophagy via the Akt/NF-κB pathway in AGS cells [22]. Pectolinarigenin, isolated from *Cirsium chanroenicum*, displayed anticancer activity through autophagy induction of human gastric cancer AGS and MKN-28 cells via the downregulation of the PI3K/Akt/mTOR pathway [88]. Perillaldehyde increased AMPK phosphorylation, leading to autophagy in human gastric cancer MFCs mouse and GC9811-P cells [89]. However, quercetin activated autophagy protection against the apoptosis in AGS and MKN-28 gastric cancer cells, which signified that autophagy might have contributed to the survival of cancer cells [90]. Therefore, autophagy induction by natural bioactive compounds might possibly be targeted as a potential therapeutic approach to control gastric cancer. 

## 5. Role of Bioactive Natural Compounds to Arrest Cell Cycle in Gastric Cancer 

The cell cycle is regulated through a series of control systems that in turn promote or inhibit cell division. Programmed cell death and cell cycle regulation occur together in many cancerous cells, since the tumor suppressor gene p53 and downstream proteins regulate both events [91]. A variety of natural bioactive components were described as causing cell death and inhibited cell proliferation by seizing the cell cycle according to the phase of cell cycle arrest (Table 3). Berberine, a traditional Chinese medicine normally used for gastroenteritis, inhibited proliferation of SGC-7901 gastric cancer cells in addition to inducing G1 arrest in the cell cycle phase and activated apoptosis [35]. Toosendanin, a triterpenoid, increased the proportion of cells in the G1 and S phase by activation of β-catenin signaling in gastric carcinoma [60,92]. Moreover, ginsenoside-Rh2 inhibited proliferation of SGC-7901 side population gastric cancer cells by the induction of cell cycle arrest, as well as cell apoptosis, and altered BAX/Bcl-2 protein expression [38]. Crosolic acid, isolated from *Actinidia valvata* Dunn. *Radix*, increased the sub G1 population of the cell cycle and decreased p65, bcl-2, Fas, and smac mRNA expression, and increased IκBα, bax, and survivin mRNA expression, which induced apoptosis of the human gastric cancer cell line BGC823 through down-regulation of the NF-κB pathway [54]. It has been found that rottlerin suppressed cell growth, induced autophagy as well as apoptosis, and reduced migration in addition to invasion in SGC-7901 and MGC-803 GC gastric cancer cells through mTOR and S-phase kinase-associated protein 2 downregulation [83]. Additionally, deacetylisovaltratum, a traditional Chinese herbal medicine *Patrinia heterophylla* Bunge, inhibited the cell viability of AGS and HGC-27 cells and induced G2/M cell cycle arrest via disruption of mitochondrial membrane potential as well as induction of caspase-dependent apoptosis [55].

## 6. Anti-Angiogenesis Effects of Natural Bioactive Products in Gastric Cancer

Angiogenesis is the most common pathway for new vessel formation in cancer [93]. Anti-angiogenic agents were studied and developed for anti-cancer therapies because angiogenesis can cause tumor growth [94]. The vascular endothelial growth factor (VEGF) signaling pathway plays an essential role in regulating tumor angiogenesis, which can be used as a therapeutic target in numerous types of human gastric cancers [95]. Inhibition of VEGF leads to anti-angiogenesis in various animal and cell line models [96]. VEGFs have an important role in forming new blood vessels, including angiogenesis and vasculogenesis (Figure 3). A dietary flavonoid, luteolin, has been found to prevent angiogenesis in gastric cancer cells of MGC-803 and Hs-746T via the suppression of Notch1)/VEGF signaling [22]. Cyperenoic acid, a sesquiterpene isolated from *Croton crassifolius*, reduced vascular endothelial growth factor A (Vegfa or VEGF-A) genes by targeting the Vegfa-Kdr and Angpt-Tie signaling pathways [97]. Moreover, zerumbone, a bioactive component of ginger, showed anti-angiogenesis activity in AGS cells by reducing VEGF expression and inhibiting NF-κB [98]. Plumbagin inhibits tumor angiogenesis of gastric carcinoma via reduction of VEGF, VEGRF2, and MVD expression in gastric carcinoma in mice by the modulating nuclear factor-kappa B pathway [99]. Moreover, nitidine chloride, *Zanthoxylum nitidum* (Roxb) DC, was found to inhibit the signal transducer as well as activator of transcription 3 (STAT3) signaling in SGC-7901 and AGS human gastric cancer cell lines, which is related to tumor angiogenesis [100]. Additionally, treatment of nitidine chloride decreased the tumor volume through angiogenesis inhibition via reduction of STAT3 and VEGF levels in a xenograft mouse model induced by SGC-7901 cells [100]. Therefore, natural bioactive compound can effectively use certain VEGF subtypes, including VEGFA156, VEGFA121, VEGFR1, and VEGFR2, for the treatment of gastric cancer.

**Table 3 cancers-13-04502-t003:** Cell cycle arrest by bioactive food components in gastric cancer. (↑ increase, ↓ decrease).

Phase of Cell Cycle Arrest	Classification	Compound	Source	Experimental Model	Dose; Duration	Mechanism	References
G0/G1	Alkaloids	Berberine	(family: Ranunculaceae) *Coptidis japonica* Makino *Rhizoma*	SGC-7901	5, 10, 20 µM; 24, 48 h		[35]
G0/G1	Phytosterols	Ginsenoside-Rh2	(family: Araliaceae) *Panax ginseng* C.A. Mey *Radix*	SGC-7901	5, 10, 20 μg/mL; 24, 48 h	↑Bax ↓Bcl-2	[38]
G0/G1	Terpenoids	Crosolic acid	(family: Actinidiaceae) *Actinidia valvata* Dunn *Radix*	BGC-823	20, 40, 80 μg/mL; 72 h	↑Bax, smac, IκBα ↓Fas, Bcl-2, p65, p-IκBα, NF-κB	[54]
G1	Polyphenols	Rottlerin	(family: Euphorbiaceae) *Mallotus philipensis* Muell.	SGC-7901, MGC-803	2, 4, 8 µM; 24 h	↑LC3-II ↓mTOR, Skp2	[83]
G1/S	Terpenoids	Toosendanin	(family: Meliaceae) *Melia toosendan* Sieb et Zucc *Cortex et Fructus*	(1) AGS (2) HGC-27	(1) 0.5, 1, 2 μM; 48 h (2) 0.5, 1, 2 μM; 36 h	↑c-caspase-3, -8, -9, c-PARP, Bax, p-p38 ↓Bcl-2, Bcl-xL, Mcl-1, survivin, XIAP	[92]
S	Alkaloids	Cyclovirobuxine D	(family: Buxaceae) *Buxus microphylla* Richardii *Radix*	MGC-803, MKN-28	30, 60, 120 μM/L; 48 h	↑c-caspase-3, Bax ↓Bcl-2	[36]
S	Alkaloids	GFG-3a	(family: Meripilaceae) *Grifola frondose* (Diks.) Gray *Mycelia*	SGC-7901	100, 200 μg/mL; 24, 48 h	↑RBBP4, caspase-3, -8, p53, Bax, Bad ↓RUVBL, NPM, Bcl-2, Bcl-xL, PI3K, Akt1	[64]
G2/M	Flavonoids	Trifolirhizin	(family: Fabaceae) *Sophora flavescens* Aiton. *Radix*	MKN-45	20, 30, 40 µg/mL; 48 h	↑caspase-9, -3, c-PARP, p53, p38 ↓EGFR, CDC2, cyclin B, ΔΨm	[37]
G2/M	Phytosterols	Tanshinone IIA	(family: Lamiaceae) *Salviae miltiorrhiza* Bunge. *Radix*	AGS	2.0, 3.7, 5.5 µg/mL; 24, 48 h	↑TNF-α, Fas, p-p38, p-JNK, p53, p21, caspase-8, -3 ↓p-ERK, CDC2, cyclin A, cyclin B1	[39]
G2/M	Terpenoids	Deacetylisovaltratum	(family: Caprifoliaceae) *Patrinia heterophylla* Bunge.	(1) AGS (2) HGC-27	(1) 4, 8, 16 μM; 24 h (2) 10, 20, 30 μM; 24 h	↑p21, caspase-3, c-PARP ↓p-STAT3, pro-caspase-9, ΔΨm	[55]

## 7. Anti-Metastasis Effects of Bioactive Compounds in Gastric Cancer

Metastasis is a major contributor of death in cancer patients, arising from a growing tumor from which cells escape to distant organs of body [101]. Targeting metastasis is an attractive strategy in cancer treatment. Anti-metastatic ability is highlighted in diverse natural bioactive products in vitro and in vivo models. which are described below. Sulforaphane, an organosulfur compound isolated from *Brassica oleracea* var. *italica* Plenk (Brassicaceae), exerted anti-metastatic ability on AGS and MKN-45 cells [102]. Isoliquiritigenin, a phenol found in *Glycyrrhiza glabra* (Fabaceae), inhibited tumor migration and metastasis on MKN-28 cells [103]. Dehydroeffusol, a benzenoid derived from *Juncus effusus* L. *Radix et Medulla* (Juncaceae), inhibited matrix metalloproteinase 2 (MMP-2) and VE-cadherin expression, resulting in reduction of the cell-to-cell adherent junction in AGS and SGC-7901 cells [104]. Baicalein, a well-known flavone found in the roots of *Scutellaria baicalensis* Georgi *Radix* (Lamiaceae), restrains motility, migration, and invasion of AGS gastric cancer cells via downregulation of N-cadherin, vimentin, ZEB1, ZEB2, and TGF-β/Smad4 [105]. Andrographolide, a labdane diterpenoid from the herb *Andrographis paniculata* Nees *Herba* (Acanthaceae), inhibits proliferation and metastasis of gastric cancer SGC-7901 via cell cycle arrest; upregulation of Bax, Bik, and TIMP-1/2; and downregulation of Bcl-2, CD147, MMP-2, and MMP-9 [106]. Blockages of tumor proliferation and metastasis of several bioactive compounds are presented in Table 4 and Figure 4. It has been found that evodiamine, isolated from *Evoida rutaecarpa* (Rutaceae), suppressed the epithelial–mesenchymal transition (EMT) of AGS and SGC-7901 gastric cancer cells via inhibition of the Wnt/β-catenin signaling pathway [107]. A triterpenoid found from *Melia toosendan* Sieb et Zucc (Meiliaceae), named toosendanin, has anti-metastatic capability on SGC-7901 cells through inhibition of the epithelial–mesenchymal transition of gastric cancer by upregulating miR-200a and e-cadherin and suppressing β-catenin [60]. Low-molecular-weight citrus pectin (LCP), derived from tangerines, grapefruits, lemons, and oranges, demonstrated anti-metastatic effects by treatment on AGS cells [108]. N-butylidenephthalide inhibited tumor metastasis in AGS, NCI-N87, and TSGH-9201 cells. The compound promoted e-cadherin expression while downregulating n-cadherin and vimentin slug. The activity of e-cadherin was repressed on the other hand, which inhibited EGFR kinase activity [57]. The mechanism leads to downstream regulation of multiple growth factor-related activities, which is associated with anti-metastatic activities of such natural bioactive products. In other aspects, the Bcl-2 family of proteins was also found to play a role in anti-metastatic effects of natural bioactive products [109]. Many other factors including PI3K, Akt, Rac1, and CDX1/2 play a role in anti-metastatic activity of natural bioactive compounds, some of which are also related to apoptosis of tumor cells. As it is unclear whether natural products exert anti-metastatic effects in a multi-target manner, further study is therefore required to distinguish the specific mechanism.

**Table 4 cancers-13-04502-t004:** Metastasis-inhibiting bioactive food components *in vitro* in gastric cancer. (↑ increase, ↓ decrease).

Classification	Compound	Source	Experimental Model	Doses	Efficacy	Mechanisms	Reference
Alkaloids	Evodiamine	(family: Rutaceae) *Tetradium ruticarpum*	AGS, SGC-7901	2 µM; 48 h	Inhibition of EMT	↓β-catenin, cyclin D1, c-Myc	[107]
Organosulfur compounds	Sulforaphane	(family: Brassicaceae) *Brassica oleracea* var. *italica* Plenk	AGS, MKN-45	31.25, 62.5, 125, 250 μg/mL; 48 h	Inhibition of metastasis	↑CDX1, CDX2	[102]
						↑miR-326, miR-9	
Polyphenols	Isoliquiritigenin	(family: Fabaceae) *Glycyrrhiza glabra Radix*	MKN-28	20 µM; 24, 48, 72 h	Inhibition of migration, invasion, Induction of apoptosis and autophagy	↓Caspase-3, Bax, Bcl-2, PI3K, Akt, mTOR	[103]
Polyphenols	Dehydroeffusol	(family: Juncaceae) *Juncus effusus* L. *Radix et Medulla*	AGS, SGC-7901	12, 24, 48 µM; 24 h	Reduction of cell–cell adherent junction	↓VE-cadherin, MMP-2	[104]
Polyphenols	Paeonol	(family: Paeoniaceae) *Paeonia suffruticosa* Andr. *Cortex*, (family: Asclepiadaceae) *Cynanchum paniculatum* K. Schum *Radix*	BGC-823	0.1, 0.2, 0.4 mg/mL; 24, 48 h	Inhibition of proliferation, invasion, and migration, Induction of apoptosis	↓MMP-2, MMP-9	[58]
Polyphenols	Baicalein	(Lamiaceae) *Scutellaria baicalensis* Georgi *Radix*	AGS	25, 50 µM; 24 h	Inhibition of motility, migration, invasion	↓N-cadherin, vimentin, ZEB1, ZEB2, TGF-β/Smad4	[105]
Terpenoids	Andrographolide	(family: Acanthaceae) *Andrographis paniculata* Nees *Herba*	SGC-7901	5, 20, 40 µg/mL; 24, 48, 72 h	Inhibition of proliferation, invasion, metastasis	↑Bax, Bik, TIMP-1/2, ↓Bcl-2, CD147, MMP-2, MMP-9, survivin	[106]
Terpenoids	Toosendanin	(family: Meliaceae) *Melia toosendan Sieb* et Zucc *Cortex et Fructus*	SGC-7901	0.5, 1 µM; 48 h	Inhibition of invasion, migration, EMTInduction of apoptosis and cell cycle arrest	↑E-cadherin ↓β-catenin	[60]
						↑miR-200a	

## 8. Chemotherapy Resistance and Natural Bioactive Products in Gastric Cancer

Drug resistance is an important issue in cancer treatment and is known as a primary factor limiting cancer treatment [110]. Several studies have indicated that natural bioactive compounds could be used along with the primary drug to overcome drug resistance and reinforce its efficacy. In vitro drug resistance-overcoming bioactive food components in gastric cancer and their target signals are presented in Figure 5. Isorhamnetin, a flavonoid metabolite of quercetin commonly found in onions, minimized the apoptotic effects of capecitabine via inhibition of NF-κB and various NF-κB regulated gene products in tumor cells [111]. Liquiritin, isolated from *Glycyrrhiza uralensis* Fischer. *Radix* (Leguminosae/Fabaceae/Fabaceae), could circumvent the resistance of cisplatin-based chemotherapy via suppression of cell proliferation and induce apoptosis, autophagy, and G0/G1 phase cell cycle arrest against DDP-resistant gastric cancer cells [112]. Astragalus polysaccharide and apatinib co-treatment were reported to enhance apoptosis compared to apatinib monotherapy [113]. The efficacy of astragalus polysaccharide, an active component derived from *Astragalus mambranaceus* Bunge *Radix* (Leguminosae/Fabaceae/Fabaceae), arises mainly from its ability to inhibit autophagy of apatinib-resistant cells, which serves as a survival mechanism. Tanshinone IIA solution combined with doxorubicin showed anticancer effects against doxorubicin-resistant cell lines, including SNU-638, SNU-668, SNU-216, and SNU-620 [114]. Apoptosis was mainly induced by inhibition of multidrug resistance-associated protein 1 (MRP1). Although specific targets vary, most natural bioactive compounds aim to prevent drug resistance by downregulating Akt and NF-κB and following pathways (Figure 5). Mineral isorhamnetin from quercetin inhibited cell viability and prevented drug resistance by downregulating NF-κB. Liquirtin from the *Glycyrrhiza* genus promoted p53 and p21 and caspase cleavages while inhibiting cyclin activities. The compound’s anti-resistant ability may be focused on apoptotic effects. Other factors such as Bax/Bcl-2 in mitochondria, and ERK1/2, MMP2, and PARP are broadly affected by many natural bioactive compounds. 

## 9. Limitation and Future Perspectives of Natural Bioactive Food Products in Gastric Cancer Treatments

Gastric cancer is known to account for the fifth highest incidence and the fourth highest mortality among all cancers worldwide [1]. Chemotherapy is one of the methods typically used in advanced gastric cancer treatment, but it exerts severe side effects that limit the efficacies and decrease quality of life. Development of therapeutic remedies with less adverse effects and lower chemo-resistance is required. Natural bioactive food products are emerging as alternative resources to combat gastric carcinoma. Therefore, several natural bioactive resources obtained from dietary fruits and vegetables were discussed. Curcumin and oligosaccharide isolated from tomato, sulforaphane derived from broccoli, and citrus pectin originated from tangerine, grapefruit, lemon, and orange are good examples. These medicinal resources are still being extensively used in traditional medicine. Many natural bioactive food products were shown to exhibit multiple effects. The variety is attributed to the structural diversity and multi-target characteristic of natural compounds [115]. Additionally, clinical trials were excluded to focus on laboratory experiments highlighting specific biological pathways. Several investigations were insufficient to elucidate anti-cancer mechanisms at molecular levels in gastric cancer. They were generally focused on the cytotoxicity of the chemicals or the reporting of newly discovered compounds, which makes incisive research burdensome. By and large, more than half of the studies only carried out experiments in vitro. More in vivo studies are recommended to bridge the advance to clinical trials and therapeutic use.

Natural bioactive food products are indeed effective in the single compound to single target mechanistic perspective; however, it is worth highlighting the complex interactions between many compounds. While the importance of studying the interactions between multi-compound natural bioactive food products and other drugs was previously highlighted in many literatures, it is also important to further investigate the interactions between different natural bioactive food products, including herbal medicines, in a biochemical manner [116]. A systemic approach with a focus on structural similarities of several phytochemical compounds and human metabolites is a potential way of clearly highlighting the efficacies of multi compound drugs. Despite the value of natural bioactive food products as medicinal agents, it is important that users as well prescribers be aware of the potentially cross-reactivity and toxicity of natural bioactive food products. Indeed, it has often been stated that natural bioactive products are toxins that are taken at lower therapeutic doses. To avoid this problem, it is required to modify the natural chemical. Therefore, it is important to recognize that unmodified natural bioactive food products may have suboptimal efficacy or absorption, distribution, metabolism, excretion, as well as toxicity (ADMET) properties. Thus, for development of natural bioactive food products that lead to successful drugs, chemical modifications or combinations with other compounds are highly required. Furthermore, clinical development requires a sustainable and suitably economically viable compound supply with sufficient quantities of natural bioactive food products.

## 10. Conclusions

In this review, we summed up several natural bioactive food products that have anti-cancer efficacy against gastric cancer. Several epidemiological investigations have been recommended, namely that the consumption of bioactive dietary food products such as spices, vegetables, fruits, roots, bulk, and leafs are inversely related to the risk and control of gastric cancer. In vitro and in vivo studies have been exposed, namely that dietary bioactive products mainly induced cell death by apoptosis and autophagy, cell cycle arrest, inhibition of angiogenesis and metastasis, and circumvention of chemo-resistance against stomach cancer cells through various molecular mechanisms. Several compounds showed multiple efficacies, attributed to structural complexity and multiple target pathways and proteins of bioactive dietary food products. Thus, natural substances implicate possibilities of being used in nutrition or medications, which may lead to novel discoveries in alternative medicine in cancer treatment. Additionally, attention should be paid to the bioavailability and safety of dietary food product consumption and a promising approach for the management and prevention of gastric cancer. This review provides data for future research and clinical trials to develop novel drugs from natural bioactive food products for gastric cancer treatment.

## Figures and Tables

**Figure 1 cancers-13-04502-f001:**
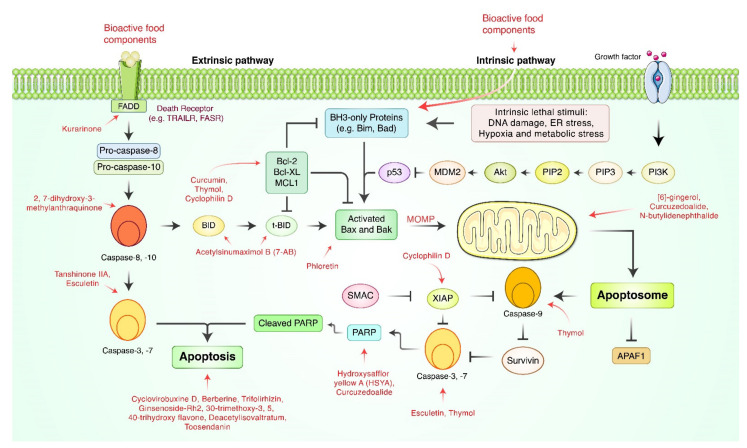
Schematic diagram of natural bioactive food product-mediated apoptosis signaling pathways. FADD, Fas-associated proteins with death domain; TRAILR, TNF-related apoptosis-including ligand receptor; FASR, Fas receptor; tBid, truncated Bid; PARP, poly ADP-ribose polymerase; APAF1, apoptotic protase activating factor 1; MOMP, mitochondrial outer membrane permeabilization; PIP2, phosphatidylinositol-3,4-bisphosphate; PIP3, phosphatidylinositol-3,4,5-triphosphate; PI3K, phosphoinositide 3-kinase.

**Figure 2 cancers-13-04502-f002:**
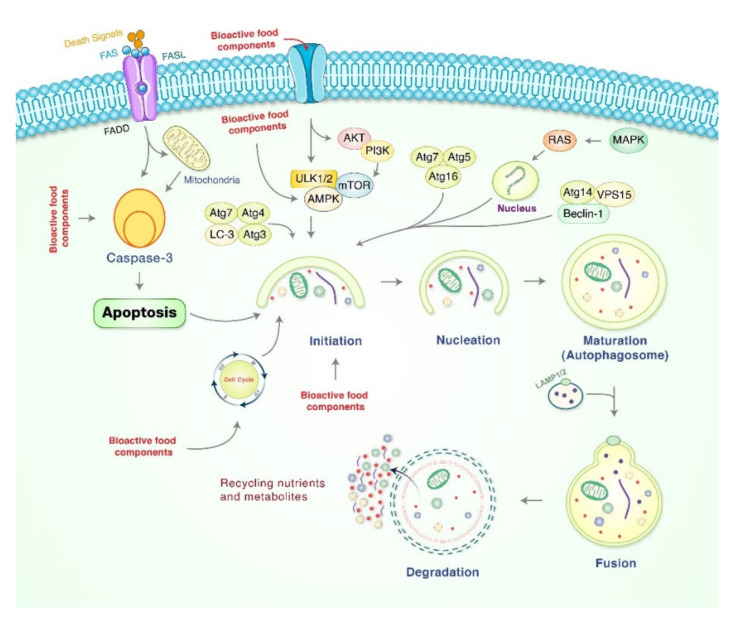
Bioactive compounds regulate molecular mechanisms of autophagy. Bioactive compounds initiate autophagy by the formation of a pre-autophagosomal structure via association of PI3K-AMPK, mammalian target of rapamycin (mTOR), ULK1, Vps34, and the Beclin-1 complex, which contribute to the formation of the pre-autophagosomal structure in addition to activating phagophore formation. Fusion of mature autophagosome as well as lysosome causes autolysosome formation. Lastly, elimination of molecules happens by acid hydrolases, which produce nutrients and recycle metabolites.

**Figure 3 cancers-13-04502-f003:**
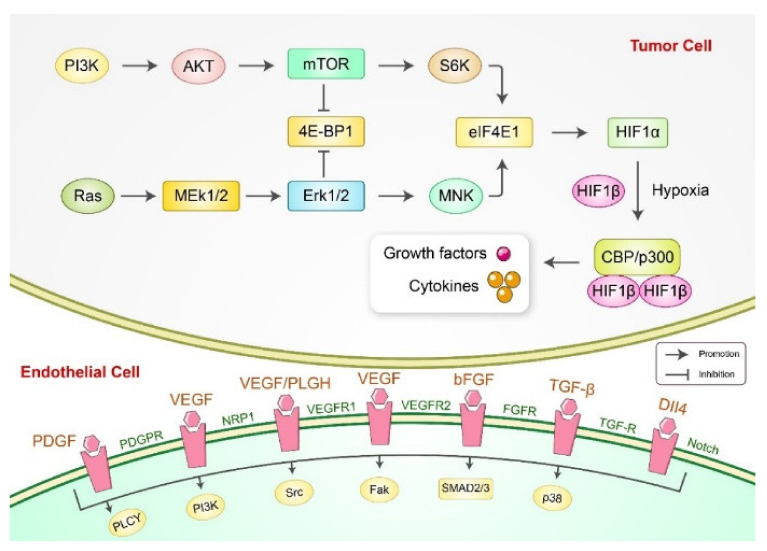
Schematic diagram of angiogenesis signaling pathways. PI3K, phosphoinositide 3-kinase; Akt, protein kinase B; mTOR, mammalian target of rapamycin; S6K, S6 kinase; MEK1/2, mitogen-activated protein kinase kinase 1/2; ERK1/2, extracellular signal-regulated kinase 1/2; MNK, mitogen-activated protein kinase-interacting kinase; 4E-BP1, eIF4E-binding protein 1; elF4E1, eukaryotic initiation factor 4E 1; HIF-1α, hypoxia-inducible factor-1 alpha; HIF-1β, hypoxia-inducible factor-1 beta; CBP, CREB-binding protein; p300, CBP homolog; PDGF, platelet-derived growth factor; PDGFR, platelet-derived growth factor receptor; VEGF, vascular endothelial growth factor; NRP1, neuropilin-1; PlGF, placental growth factor; VEGFR-1, vascular endothelial growth factor receptor-1; VEGFR-2, vascular endothelial growth factor receptor-2; bFGF, basic fibroblast growth factor; FGFR, fibroblast growth factor receptors; TGF-β, transforming growth factor beta; TGF-R, transforming growth factor receptor; Dll4, delta-like ligands.

**Figure 4 cancers-13-04502-f004:**
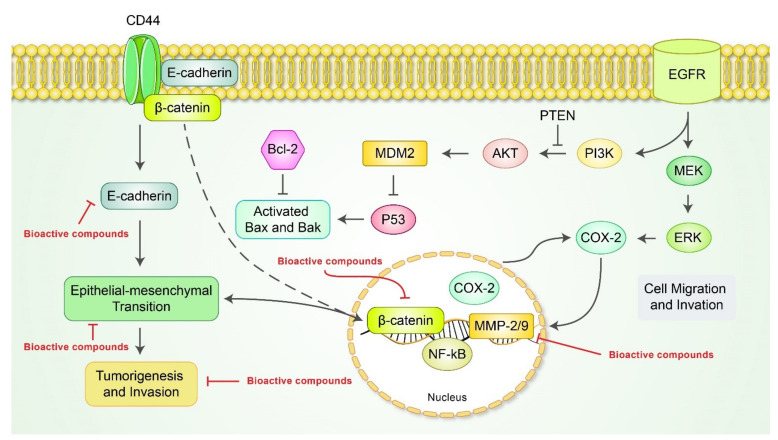
Schematic diagram of metastasis signaling pathways and regulation by bioactive compounds. Akt, protein kinase B; Bak, Bcl-2 antagonist/killer 1; Bax, Bcl-2-like protein 4; Bcl-2, B-cell lymphoma 2; CD44, homing cell adhesion molecule; COX-2, cyclooxygenase 2; EGFR, epidermal growth factor receptor; ERK, extracellular signal-regulated kinase; MDM2, murine double minute 2; MEK, matrix metalloproteinase-2/9; NF-κB, nuclear factor kappa-B; PI3K, phosphoinositide 3-kinase; PTEN, phosphatase and tensin homolog.

**Figure 5 cancers-13-04502-f005:**
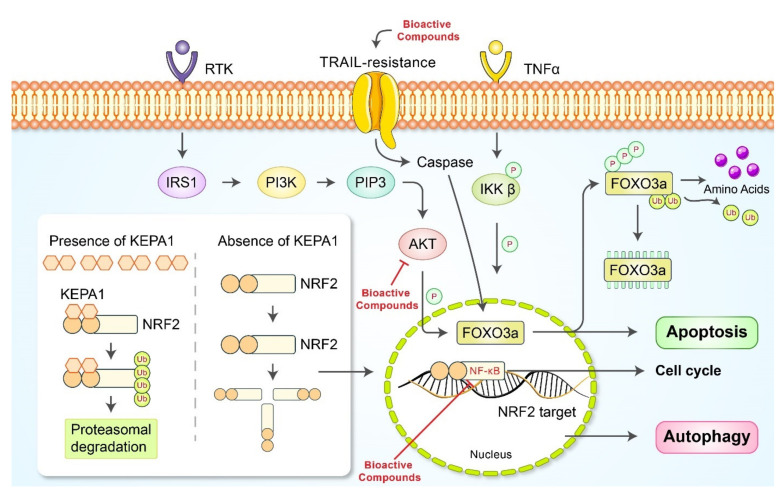
Schematic diagram of resistance signaling pathway. RTK, receptor tyrosine kinase; IRS1, insulin receptor substrate 1; PI3K, phosphoinositide 3-kinases; PIP3, phosphatidylinositol (3,4,5)-trisphosphate; AKT, protein kinase B (PKB); FOXO3a, forkhead box O 3; IKK-β, inhibitor of nuclear factor κB kinase subunit beta; TNF-α, tumor necrosis factor α; Ub, ubiquitin; KEAP1, Kelch-like ECH-associated protein 1; NRF2, nuclear factor erythroid 2-related factor 2.

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
