# Peer review of "Potential of Bioactive Food Components against Gastric Cancer: Insights into Molecular Mechanism and Therapeutic Targets"

_cancers, 2021, doi:10.3390/cancers13184502_

Round 1

Reviewer 1 Report

Comments to the Authors:

The authors of the article entitled "Emerging Potential of Naturally Occurring Molecules against Gastric Cancer: Insights Molecular Mechanism and Therapeutic Targets" discussed the potential and usefulness of different naturally occurring molecules, compounds or extracts as an aid in treatment of gastric cancer.

Comments:

  1. Title: The title is not clearly written, it appears there is something missing in the second part of the title, possibly “….. Insights into Molecular Mechanism and Therapeutic Targets
  2. Abstract, lines 18,19: delete “been” from the sentence “Accumulated evidences and epidemiological studies have been indicated that …
  3. Abstract, line 20: delete “been”
  4. Abstract, lines 23-25: the sentence is not clear “Although chemotherapy remains the standard treatment for advanced gastric cancer along with surgery, radiation therapy, hormone therapy and immunotherapy, but its adverse side effects including neutropenia, stomatitis, mucositis, diarrhea, nausea, and emesis are well documented.” Consider omitting “although” and replacing “but” with “and”.
  5. Abstract, line 26: The meaning of this sentence is not clear: Additionally, intake of naturally occurring phytochemicals could increase the efficacy of gastric chemotherapy and chemotherapeutics resistance”. Did the authors mean that phytochemicals increase both the efficacy and resistance at the same time?
  6. Page 2, line 53 : the sentence “Exposure to unremovable toxins, trauma, or infection can occur mutagenic chronic inflammatory response, leading to dyplasia [12].” Is not clear, consider rephrasing. Could you also define “unremovable toxins”?
  7. Page 2, line 77: The title should be rephrased, it is not clear: Role of naturally occurring molecules in gastric cancer managements by apoptosis. Did the authors mean: Role of naturally occurring molecules affecting apoptosis and their influence on gastric cancer management?
  8. Section 2.1. presents a lot of information, could the authors format the chapter so that it includes more paragraphs and presents this large amount of information in more organized manner? Also, throughout of this section, the authors presented information focusing more on the inhibition of proliferation or tumour growth, it would be easier for the reader, if these sections would be collected in separate section(s).
  9. Page 14, line 269: delete “have been” from “Several have been studies demonstrated…”
  10. Section 2.1. similar comment as Comment 8 – could this section be separated into two subsections, one more focusing on direct effect of extract on apoptosis and the second one focusing on tumour growth inhibition?
  11. At the end of the manuscript, there are incorrectly numbered pages – after the Table 14, there is a page numbered 3 of 52, therefore, I will refer to lines in this comment. The text between lines 739 – 750 is describing the methodology/systemization of this review and could be moved before section 2 as new section, as it explains the rationale for organizing the sections, described in the main text.
  12. Finally, the article is very interesting and informative, and is, in general, well-written, however I would suggest that the authors thoroughly check the manuscript for some minor language polishing, word choice, and minor grammatical errors.

Author Response

The authors of the article entitled "Emerging Potential of Naturally Occurring Molecules against Gastric Cancer: Insights Molecular Mechanism and Therapeutic Targets" discussed the potential and usefulness of different naturally occurring molecules, compounds or extracts as an aid in treatment of gastric cancer.

>> (Response) First of all, we would like to express our sincere gratitude for the time and effort the reviewer had put into reviewing our manuscript.

Comments:

  1. Title: The title is not clearly written, it appears there is something missing in the second part of the title, possibly “….. Insights into Molecular Mechanism and Therapeutic Targets

>> (Response) we modified the title. New title is: Emerging Potential of Naturally Occurring Molecules against Gastric Cancer: Insights Pharmacological targets and Molecular Mechanisms (page 1, line 2).

  1. Abstract, lines 18,19: delete “been” from the sentence “Accumulated evidences and epidemiological studies have been indicated that …

>> (Response) We deleted ‘been’ from this sentence (page 1, line 28).

  1. Abstract, line 20: delete “been”

>> (Response) We deleted ‘been’ from this sentence (page 1, line 30).

  1. Abstract, lines 23-25: the sentence is not clear “Although chemotherapy remains the standard treatment for advanced gastric cancer along with surgery, radiation therapy, hormone therapy and immunotherapy, but its adverse side effects including neutropenia, stomatitis, mucositis, diarrhea, nausea, and emesis are well documented.” Consider omitting “although” and replacing “but” with “and”.

>> (Response) We modified the sentence according to the reviewer suggestion (page 1, line 33-35).

  1. Abstract, line 26: The meaning of this sentence is not clear: Additionally, intake of naturally occurring phytochemicals could increase the efficacy of gastric chemotherapy and chemotherapeutics resistance”. Did the authors mean that phytochemicals increase both the efficacy and resistance at the same time?

>> (Response) We modified the sentence (page 1, line 36-37).

  1. Page 2, line 53: the sentence “Exposure to unremovable toxins, trauma, or infection can occur mutagenic chronic inflammatory response, leading to dyplasia [12].” Is not clear, consider rephrasing. Could you also define “unremovable toxins”?

>> (Response) We modified the sentence (page 2, line 65-66). Unremovable toxins means ‘toxin not able to be removed or non-releasable’

  1. Page 2, line 77: The title should be rephrased, it is not clear: Role of naturally occurring molecules in gastric cancer managements by apoptosis. Did the authors mean: Role of naturally occurring molecules affecting apoptosis and their influence on gastric cancer management?

>> (Response) We modified the title (page 3, line 103). ‘Role of naturally occurring molecules affecting apoptosis on gastric cancer managements.

  1. Section 2.1. presents a lot of information, could the authors format the chapter so that it includes more paragraphs and presents this large amount of information in more organized manner? Also, throughout of this section, the authors presented information focusing more on the inhibition of proliferation or tumour growth, it would be easier for the reader, if these sections would be collected in separate section(s).

>> (Response) Section 2.1 (now section 3.1) contain huge information regarding natural product induced apoptosis in several gastric cancer cells. As the reviewer suggestion, we separate more paragraphs with specified some key words such as compound with general apoptosis effects, MAPK-mediated effects, TRAIL-mediated effects, intrinsic Bcl-xL/Bcl-2-mediated, p53-mediated, and in vivo pathway (page 3-6).

  1. Page 14, line 269: delete “have been” from “Several have been studies demonstrated…”

>> (Response) We corrected the sentence (page 13, line 276).

  1. Section 2.1. similar comment as Comment 8 – could this section be separated into two subsections, one more focusing on direct effect of extract on apoptosis and the second one focusing on tumour growth inhibition?

>> (Response)We divided two section in section 2.1 (now 3.1) according to the reviewer suggestion.

  1. At the end of the manuscript, there are incorrectly numbered pages – after the Table 14, there is a page numbered 3 of 52, therefore, I will refer to lines in this comment. The text between lines 739 – 750 is describing the methodology/systemization of this review and could be moved before section 2 as new section, as it explains the rationale for organizing the sections, described in the main text.

>> (Response) We fixed the page problem during the revision process. Text between lines 739 – 750 is now describing in the methodology of the manuscript (page 2, line 88).

  1. Finally, the article is very interesting and informative, and is, in general, well-written, however I would suggest that the authors thoroughly check the manuscript for some minor language polishing, word choice, and minor grammatical errors.

>> (Response) We are grateful to reviewer for the positive complement. We massively reviewed our manuscript to find out language mistake, grammatical errors, and others things.

Reviewer 2 Report

This is a very comprehensive and lengthy review, and a valuable contribution to the literature.  The figures showing the proposed point of action for the various compounds were excellent, and a wonderful resource in their own right.  The Tables were also an excellent resource.

Overall, I think the paper could be reduced in length. There was too much minor detail included throughout the text (eg. line 164-165: The apoptotic AGS cells increased from 1.25% in control to 46.3% at ....; line 113-114: They were combined at a ratio of 1:4, 20 and 80uM each). 

The division into compounds affected different biological processes was good, although within each section more sub-headings and/or separate paragraphs should be included, since the existing large blocks of text are difficult to read. 

The section on metastasis could examine more carefully whether the published data really was relevant to metastasis. It wasn't always clear how the conclusion was made that the compound affected metastasis.  Inhibition of metastasis can really only be demonstrated using whole animals. Scratch/transwell assays etc are approximate measures of motility and the potential for invasion, which is distinct from metastasis. Emphasis should therefore be placed on studies with animal data. Evidence for compounds with an impact on the EMT could also be separated out (although as a marker of risk of invasion, but not necessarily metastasis). 

Can Kun-Shin-Choa-Sa be explained? It is currently referred to at the end, but more introduction will be required for many readers.

Author Response

This is a very comprehensive and lengthy review, and a valuable contribution to the literature.  The figures showing the proposed point of action for the various compounds were excellent, and a wonderful resource in their own right.  The Tables were also an excellent resource.

>> (Response) First of all, we would like to express our sincere gratitude for the time and effort the reviewer had put into reviewing our manuscript. We are grateful to reviewer for the positive complement of our manuscript.

Overall, I think the paper could be reduced in length. There was too much minor detail included throughout the text (eg. line 164-165: The apoptotic AGS cells increased from 1.25% in control to 46.3% at ....; line 113-114: They were combined at a ratio of 1:4, 20 and 80uM each). 

>> (Response) We try to reduce the length of the manuscript. Initial length: 17763, now 16386.

In line 164-165, the apoptotic AGS cells increased from 1.25% in control to 46.3% at ...., we modified the sentence (page 5, line 179).

In line 113-114, we modified the sentence (page 3, line 137).

The division into compounds affected different biological processes was good, although within each section more sub-headings and/or separate paragraphs should be included, since the existing large blocks of text are difficult to read. 

>> (Response) As the reviewer suggestion, we separated more paragraphs with specified some key words through the manuscript which we marked by blue color.

The section on metastasis could examine more carefully whether the published data really was relevant to metastasis. It wasn't always clear how the conclusion was made that the compound affected metastasis.  Inhibition of metastasis can really only be demonstrated using whole animals. Scratch/transwell assays etc are approximate measures of motility and the potential for invasion, which is distinct from metastasis. Emphasis should therefore be placed on studies with animal data. Evidence for compounds with an impact on the EMT could also be separated out (although as a marker of risk of invasion, but not necessarily metastasis). 

>> (Response) We massively revised and deleted some unnecessary sentence from this section. Additionally, we separated compounds with an impact on the EMT and more paragraphs with relevant information (page 30-31).

Can Kun-Shin-Choa-Sa be explained? It is currently referred to at the end, but more introduction will be required for many readers.

>> (Response) Kun-Shin-Choa-Sa is not highly related with this study. Therefore, we deleted this information from our manuscript.

Reviewer 3 Report

Emerging Potential of Naturally Occurring Molecules against Gastric Cancer: Insights Molecular Mechanism and Therapeutic Targets by Seog Young Kang et al is a review article about natural compounds and mixtures that might have the potential to inhibit Gastric cancer-related targets. This is a very relevant work for the Cancers journal community. I have the following comments/questions:

The following sentence is taken from the Abstracts section, “However, natural product structural stability and powerful bioactivity are important to develop novel treatments for gastric cancer that may minimize such adverse effects.”, claim that natural products have structural stability and possess powerful bioactivity. It is not clear how the authors came to this conclusion.

Section 2.3 mentions how the Jinlong capsule (JLC) was used to treat GC. Jinlong contains many chemical components, and it is not clear how one could assign a mechanism of action to this decoction.

Authors should add the following Nature paper [Ref1; see below], to their references and consider addressing some issues associated with natural products in becoming a drug. This article (Ref1) elegantly addresses some of these issues, and I quote few relevant lines from the article here, “it is important to acknowledge that unmodified NPs may possess suboptimal efficacy or absorption, distribution, metabolism, excretion and toxicity (ADMET) properties. So, for development of NP hits into leads and ultimately into successful drugs, chemical modification may be required. In addition, bringing a compound into clinical development requires a sustainable and economically viable supply of sufficient quantities of the compound.”

Natural compounds as the Authors claim have diverse structures with a wide range of biological functions. This could lead to cross-reactivity and toxicity. Unfortunately, there is very little information -based on the review- on the effect of toxicity of these natural compounds.

Ref1: Natural products in drug discovery: advances and opportunities, Nature Reviews Drug Discovery vol 20, pages 200–216 (2021)

Minor issues/typos:

  • Section 2.2, page 14: Please rewrite the following sentence, “Several have been studies demonstrated that natural product extracts…”.
  • Section 2.2 is one long paragraph. To improve readability, please consider breaking the paragraph into smaller sub-paragraphs. Please also take a look at Section 6, the first paragraph is more than one page.

Author Response

Emerging Potential of Naturally Occurring Molecules against Gastric Cancer: Insights Molecular Mechanism and Therapeutic Targets by Seog Young Kang et al is a review article about natural compounds and mixtures that might have the potential to inhibit Gastric cancer-related targets. This is a very relevant work for the Cancers journal community. I have the following comments/questions:

>> (Response) First of all, we would like to express our sincere gratitude for the time and effort the reviewer had put into reviewing our manuscript.

The following sentence is taken from the Abstracts section, “However, natural product structural stability and powerful bioactivity are important to develop novel treatments for gastric cancer that may minimize such adverse effects.”, claim that natural products have structural stability and possess powerful bioactivity. It is not clear how the authors came to this conclusion.

>> (Response) We modified the sentence. Due to structural stability, potential bioavailability, and powerful bioactivity of natural product, we may say conclude the point (page 1, line 38-40).

Section 2.3 mentions how the Jinlong capsule (JLC) was used to treat GC. Jinlong contains many chemical components, and it is not clear how one could assign a mechanism of action to this decoction.

>> (Response) As a representative traditional Chinese medicine made by modern pharmaceutical technology, Jinlong Capsule (JLC) has been used for several decades to treat liver cancer with significantly improved clinical outcomes as adjuvant therapy. This was found in Chinese patent medicine (Jinlong Capsule) for gastric cancer: Protocol for a systematic review and meta-analysis. Medicine (Baltimore) has been published by Li, J., et al. 2020. Additionally, Jinlong Capsule (JLC) inhibits proliferation and induces apoptosis in human gastric cancer cells in vivo and in vitro also published by Li, D., et al. at 2018.

Therefore, these two papers well documented how JLC might be used as a mechanism of action to treat GC.

Authors should add the following Nature paper [Ref1; see below], to their references and consider addressing some issues associated with natural products in becoming a drug. This article (Ref1) elegantly addresses some of these issues, and I quote few relevant lines from the article here, “it is important to acknowledge that unmodified NPs may possess suboptimal efficacy or absorption, distribution, metabolism, excretion and toxicity (ADMET) properties. So, for development of NP hits into leads and ultimately into successful drugs, chemical modification may be required. In addition, bringing a compound into clinical development requires a sustainable and economically viable supply of sufficient quantities of the compound.”Natural compounds as the Authors claim have diverse structures with a wide range of biological functions. This could lead to cross-reactivity and toxicity. Unfortunately, there is very little information -based on the review- on the effect of toxicity of these natural compounds.

Ref1: Natural products in drug discovery: advances and opportunities, Nature Reviews Drug Discovery vol 20, pages 200–216 (2021)

>> (Response) We are grateful to reviewer for this wonderful point. Although we did not focus on toxicity effects on NPs, here we added one more paragraph with short description of cross-reactivity and toxicity of natural products in section: ‘9. Limitation and future perspectives of natural products in gastric cancer treatments’ using this reference paper.

Minor issues/typos:

  • Section 2.2, page 14: Please rewrite the following sentence, “Several have been studies demonstrated that natural product extracts…”.

>> (Response) We modified the sentence (page 13, line 276)

  • Section 2.2 is one long paragraph. To improve readability, please consider breaking the paragraph into smaller sub-paragraphs. Please also take a look at Section 6, the first paragraph is more than one page.

>> (Response) We divided into two paragraphs in section 2.2 (now 3.2) (page 13).

We massively revised and deleted some unnecessary sentence from section 6 (now section 7). Additionally, we separated compounds with an impact on the EMT and more paragraphs with relevant information (page 34-35).

Reviewer 4 Report

The present review represents a comprehensive summarization of the main scientific so far knowledge  on the effects of natural compounds in gastric cancer-derived cells and in the treatment of gastric cancer  in animal models and in (few) studies on humans. The subject of this review is relevant and, even if other reviews has been produced in recent years with similar (but not overlapping) subject, in my opnion this work deserve partucilar attenction, since it is very well done, very well organized and succeed summarizing a very large amount of scientific literature. Moreover, it is worth highlighting that this review has the merit to focus on the biological role on each specific natural compound, giving a comprehensive information on their mechanism of action so far understood. For all these reasons, in my opnion this review deserves to be published after minor revisions that authors can read below.

Minor points:

-in figure 2 the nucleus has been drawn smaller with respect to the mitochondrion: this could be misleadind in the understanding of the figure and needs to be fixed.

-there are several grammatical and typographical errors throughout the text.

Author Response

The present review represents a comprehensive summarization of the main scientific so far knowledge on the effects of natural compounds in gastric cancer-derived cells and in the treatment of gastric cancer in animal models and in (few) studies on humans. The subject of this review is relevant and, even if other reviews has been produced in recent years with similar (but not overlapping) subject, in my opnion this work deserve partucilar attenction, since it is very well done, very well organized and succeed summarizing a very large amount of scientific literature. Moreover, it is worth highlighting that this review has the merit to focus on the biological role on each specific natural compound, giving a comprehensive information on their mechanism of action so far understood. For all these reasons, in my opnion this review deserves to be published after minor revisions that authors can read below.

>> (Response) First of all, we would like to express our sincere gratitude for the time and effort the reviewer had put into reviewing our manuscript.

Minor points:

-in figure 2 the nucleus has been drawn smaller with respect to the mitochondrion: this could be misleadind in the understanding of the figure and needs to be fixed.

>> (Response) We corrected the figure 2 (page 20)

-there are several grammatical and typographical errors throughout the text.

>> (Response) We massive checked and corrected all the grammatical and typographical errors throughout the text.

Round 2

Reviewer 2 Report

There have been a number of improvements made but I think it is still too detailed.   Line 163 is typical: . TTF induced apoptosis on SGC-7901 cells treated at the dose of 2, 4, 8, 16, 32 μg/mL 163 for 24, 48 and 72 h.  

This is really too much detail for a review. 

Can the authors confirm that all studies that purportedly looked at metastasis actually did so in animal models, and didn't just extrapolate from invasion and EMT data.

Author Response

There have been a number of improvements made but I think it is still too detailed.   Line 163 is typical: . TTF induced apoptosis on SGC-7901 cells treated at the dose of 2, 4, 8, 16, 32 μg/mL 163 for 24, 48 and 72 h. This is really too much detail for a review. 

>> (Response) First of all, we would like to express our sincere gratitude for the time and effort the reviewer had put into reviewing our manuscript. We modified the reviewer indicated sentence accordingly (page 4, line 161-165)

Additionally, we checked the manuscript and deleted unnecessary information mentioned by the reviewer and marked as blue color.

Can the authors confirm that all studies that purportedly looked at metastasis actually did so in animal models, and didn't just extrapolate from invasion and EMT data.

>> (Response) We divided section 7 into in vitro and in vivo model of ‘Anti-metastasis effects of natural products in gastric cancer’ in where we collected all in vivo related natural product in ‘7.2 Anti-metastasis effects of natural products in vivo animal model’. Although in vitro model is in the same location in ‘7.1 Anti-metastasis effects of natural products in vitro gastric cancer management’. (page 30 & 31, line 550-578)

Hope the reviewer will understand. We are grateful to the reviewer to raise this issue.